# Calcium-dependent electrostatic control of anion access to the pore of the calcium-activated chloride channel TMEM16A

Andy KM Lam[1], Raimund Dutzler[2]*

[1]Department of Biochemistry, University of Zurich, Zurich, Switzerland; [2]Department of Biochemistry, University of Zurich, Zurich, Switzerland

**Abstract** TMEM16A is a ligand-gated anion channel that is activated by intracellular $Ca^{2+}$. This channel comprises two independent pores and closely apposed $Ca^{2+}$ binding sites that are contained within each subunit of a homodimeric protein. Previously we characterized the influence of positively charged pore-lining residues on anion conduction (*Paulino et al., 2017a*). Here, we demonstrate the electrostatic control of permeation by the bound calcium ions in mouse TMEM16A using electrophysiology and Poisson-Boltzmann calculations. The currents of constitutively active mutants lose their outward rectification as a function of $Ca^{2+}$ concentration due to the alleviation of energy barriers for anion conduction. This phenomenon originates from Coulombic interactions between the bound $Ca^{2+}$ and permeating anions and thus demonstrates that an electrostatic gate imposed by the vacant binding site present in the sterically open pore, is released by $Ca^{2+}$ binding to enable an otherwise sub-conductive pore to conduct with full capacity.
DOI: https://doi.org/10.7554/eLife.39122.001

## Introduction

The calcium-activated chloride channel TMEM16A is part of a large family of membrane proteins that encompasses ion channels and lipid scramblases with a common conserved molecular architecture (*Brunner et al., 2016*; *Caputo et al., 2008*; *Picollo et al., 2015*; *Schroeder et al., 2008*; *Terashima et al., 2013*; *Whitlock and Hartzell, 2017*; *Yang et al., 2008*). With respect to their fold, the TMEM16 family is related to mechanosensitive channels of the OSCA and TMC families (*Ballesteros et al., 2018*; *Pan et al., 2018*; *Zhang et al., 2018*). TMEM16A is widely expressed and contributes to important physiological processes including the transport of chloride across epithelia and the control of electrical signal transduction in smooth muscle and certain neurons (*Huang et al., 2012*; *Oh and Jung, 2016*; *Scudieri et al., 2012*). Recent investigations have defined the structural basis for ion permeation and gating in TMEM16A and revealed features that distinguish this ion channel from other homologues working as lipid scramblases (*Brunner et al., 2014*; *Dang et al., 2017*; *Paulino et al., 2017a*; *Paulino et al., 2017b*). TMEM16A harbors two ion conduction pores, each contained within a single subunit of a homodimeric protein (*Figure 1A*). Both pores function independently and are activated by the binding of two $Ca^{2+}$ ions to a site embedded within the membrane-inserted part of each subunit close to the ion conduction path (*Jeng et al., 2016*; *Lim et al., 2016*). In the open conformation, anions access the narrow neck of an hourglass-shaped pore via water-filled vestibules from the extra- and intracellular sides and permeate through the constricted part presumably after shedding most of their coordinating water molecules (*Betto et al., 2014*; *Dang et al., 2017*; *Ni et al., 2014*; *Paulino et al., 2017a*; *Qu and Hartzell, 2000*). During this process, the energetic penalty for dehydration is surmounted by positive charges placed on both sides of the neck (*Paulino et al., 2017b*). In the apo conformation, the $Ca^{2+}$ binding site is accessible from the cytoplasm (*Paulino et al., 2017a*). The binding of $Ca^{2+}$ favors conformational changes in

*For correspondence:
dutzler@bioc.uzh.ch

Competing interests: The authors declare that no competing interests exist.

the pore-lining helix α6, which provides polar and acidic sidechains that coordinate the bound cations and thus directly couples ligand binding to pore opening (*Paulino et al., 2017a*; *Peters et al., 2018*). The subsequent closure of the aqueous access pathway to the ligand binding site buries the bound calcium ions within the transmembrane electric field, which mechanistically accounts for the observed voltage dependence of activation by modulating the binding affinity of $Ca^{2+}$ (*Arreola et al., 1996*; *Brunner et al., 2014*; *Xiao et al., 2011*). The movement of the helix relays a conformational change towards the neck to release a steric barrier, which acts as a gate in the closed conformation of the channel (*Paulino et al., 2017a*). Additionally, the bound $Ca^{2+}$ ions change the charge distribution at the wide intracellular vestibule thereby removing a second, electrostatic barrier that specifically hinders the access of anions in the ligand-free state (*Paulino et al., 2017a*). Here we have used electrophysiology and Poisson-Boltzmann calculations to characterize the electrostatic control of anion permeation by the bound $Ca^{2+}$ ions. Our results reveal a strong and favorable Coulombic interaction of permeant anions with the positively charged $Ca^{2+}$ at the intracellular pore entry. These long-range interactions underlie an electrostatic gating mechanism that acts through space and that is synergistic with the opening of a steric gate, allowing an otherwise sub-conductive pore to conduct with full capacity.

## Results

### $Ca^{2+}$ binding to the transmembrane site alleviates energy barriers for anion conduction

We have previously used a simple rate model to characterize ion conduction in TMEM16A (*Läuger, 1973*; *Paulino et al., 2017b*). In this model, a central small energy barrier originating from the diffusion of an anion across the narrow neck is sandwiched between two larger barriers resulting from the desolvation of the anion upon entering the constricted part of the channel from the aqueous vestibules located on either side of the pore (*Figure 1A*, *Figure 1—figure supplement 1A,B*). The diffusion path does not contain deep energy wells and the model does not account for saturation of the pore, which is generally consistent with the high $K_M$ for chloride conduction. This model allowed us to phenomenologically interpret the effect of mutations of positively charged residues on current-voltage (I-V) relationships. Whereas single mutations in the wide vestibules do not exert a recognizable effect on conduction, mutations at the border of the neck region result in a pronounced rectification of currents whose shape depends on the location of the altered residues (*Paulino et al., 2017b*). The rectification is a consequence of both pore occupancy and the rate of barrier crossing at the applied potential. Here we use a similar analysis to investigate the influence of the bound calcium ions on anion conduction by characterizing the properties of the mutant G644P. In this mutant, the replacement of a flexible glycine at a hinge in α6 (Gly 644, *Figure 1B*) with a rigid proline increases the potency of $Ca^{2+}$ and concomitantly results in constitutive activity (*Paulino et al., 2017a*). The basal current of this mutant is highly outwardly rectifying but it progressively loses its rectification at increasing $Ca^{2+}$ concentrations until it becomes pseudo-linear in saturating conditions (*Figure 1C*, *Figure 1—figure supplement 1C*). This differs from the instantaneous currents of WT, which are linear in the entire $Ca^{2+}$ concentration range (*Figure 1—figure supplement 2*). When analyzed with our described minimal model of ion permeation, this effect seems to originate from the alleviation of local energy barriers for anion conduction at the intracellular entrance and the center of the pore as a function of $Ca^{2+}$ concentration (*Figure 1D*). This is consistent with a change in long-range interactions affecting permeant anions entering and traversing the narrow neck region upon binding of the positively charged $Ca^{2+}$. The lowering of the energy barriers can be described by a binding isotherm with $EC_{50}$ values of 60 nM and 170 nM, and Hill coefficients of 2.1 and 1.7 at 80 mV for the inner and the central barriers respectively (*Figure 1E*). As the rate constants are not strongly coupled, the similarity of their respective $EC_{50}$'s and their consistency with data from steady-state current responses provide independent evidence for a saturable effect associated with the binding of $Ca^{2+}$ to the transmembrane site identified in the structure.

Besides G644P, we have characterized the mutant Q649A that also causes constitutive activity. Located about one helix turn below Gly 644, the residue is not directly involved in $Ca^{2+}$ coordination

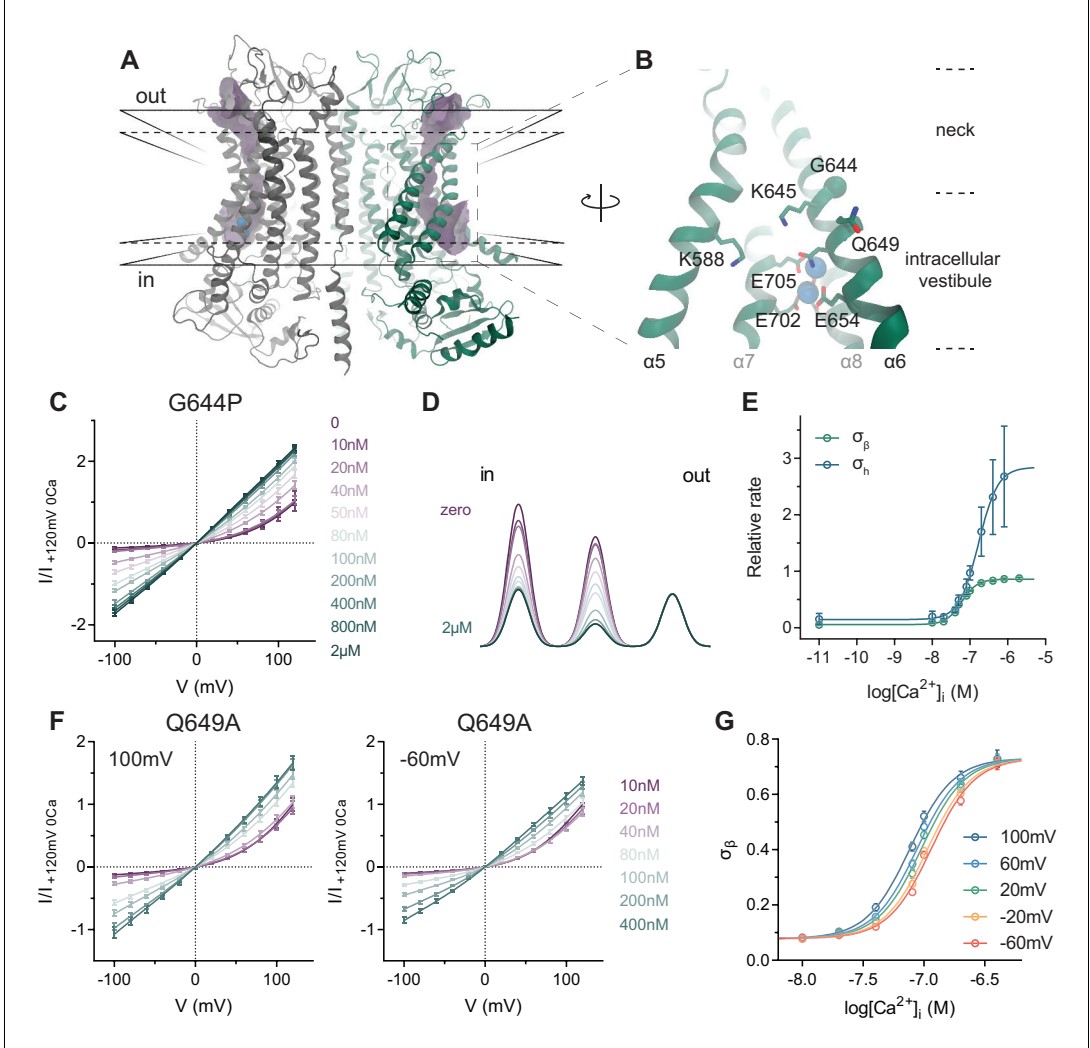

**Figure 1.** TMEM16A structure and conduction properties of constitutively active mutants. (A) Ribbon representation of the $Ca^{2+}$-bound mouse TMEM16A channel viewed from within the membrane (PDB: 5OYB), subunits are shown in unique colors. Black solid lines, outer- and innermost membrane boundaries, black dashed lines, boundaries of the hydrophobic core; blue spheres, $Ca^{2+}$ ions; violet surface, molecular surface of the pore (generated in HOLE [*Smart et al., 1996*] with a solvent probe radius of 0.7 Å). (B) Close-up of the intracellular vestibule with selected residues displayed as sticks and labelled. Green sphere, Cα of Gly 644. α3 and α4 are removed for clarity. (C) Instantaneous I-V relations of the constitutively active mutant G644P from pre-pulses at 80 mV in the presence of the indicated intracellular $Ca^{2+}$ concentrations. Solid lines are fits to *Equation 2*. Data were normalized to the fitted amplitude factor (A in *Equation 2*) and were subsequently normalized to the current amplitude at 120 mV at zero $Ca^{2+}$ (I/I $_{+120mV\ 0Ca}$). Data are mean values of normalized I-V plots from 5 to 13 individual patches, errors are s.e.m. (D) Relative energy profiles (at 0 mV) of the ion conduction path at the indicated intracellular $Ca^{2+}$ concentrations (colors as in C). Barriers are visualized as a sum of three Gaussians with the peaks and amplitudes indicating their locations and relative barrier heights respectively. (E) Relative rate of barrier crossing at the indicated location ($\sigma_h$ and $\sigma_\beta$) as a function of intracellular $Ca^{2+}$ concentration. Solid lines are fits to the Hill equation. Data are best-fit values and errors are 95% confidence intervals. (F) Instantaneous I-V relations of the mutant Q649A from pre-pulses at 100 mV (left) and −60 mV (right) in the presence of the indicated intracellular $Ca^{2+}$ concentrations. Solid lines are fits to *Equation 2*. Data were normalized as in C and are mean values of normalized I-V plots from five individual patches, errors are s.e.m. (G) Relative rate of barrier crossing at the intracellular pore entrance ($\sigma_\beta$) as a function of both intracellular $Ca^{2+}$ concentration and voltage. Solid lines are fits to the Hill equation.

DOI: https://doi.org/10.7554/eLife.39122.002

The following figure supplements are available for figure 1:

**Figure supplement 1.** Permeation model and G644P currents.

DOI: https://doi.org/10.7554/eLife.39122.003

**Figure supplement 2.** Relative contribution of open states with various $Ca^{2+}$ occupancy to the activation of WT.

DOI: https://doi.org/10.7554/eLife.39122.004

**Figure supplement 3.** Characterization of the mutant Q649A.

*Figure 1 continued*

DOI: https://doi.org/10.7554/eLife.39122.005

(*Figure 1B*). Like G644P, this mutant displays a left-shifted $EC_{50}$ and basal current that is equally outwardly rectifying (*Figure 1—figure supplement 3A*). Since the current magnitude of this mutant is larger and the kinetics of irreversible rundown is slower compared to G644P, we were able to examine the influence of the membrane potential on $Ca^{2+}$ binding by recording instantaneous currents with pre-pulses at different voltages on the same patch (*Figure 1F* and *Figure 1—figure supplement 3B—D*). At the membrane potentials tested, we observed a similar effect of $Ca^{2+}$ on the energetics of anion conduction as in G644P (*Figure 1C–F*) and as with steady-state concentration-responses, the potency of $Ca^{2+}$ is increased upon depolarization (*Figure 1F–G*, *Figure 1—figure supplement 3B–D*). Therefore, these results provide further evidence that $Ca^{2+}$ alleviates local energy barriers for anion conduction at the intracellular entrance and the middle of the pore by binding to the transmembrane site.

## The effect of the competitive antagonist $Mg^{2+}$, and the trivalent cation $Gd^{3+}$ on conduction

To confirm that the charge of $Ca^{2+}$, rather than its ability to promote gating-associated conformational changes, accounts for the observed effect, we characterized the I-V relations of G644P in the presence of the divalent cation $Mg^{2+}$ (*Figure 2*). Although $Mg^{2+}$ is incapable of activating WT even at high concentrations (*Figure 2—figure supplement 1A*), it acts as a low affinity competitive antagonist that lowers the potency of $Ca^{2+}$, indicating that it might occupy the $Ca^{2+}$ binding site (*Ni et al., 2014*). This is confirmed in the mutant G644P, for which we observed a concentration-dependent reduction of the outward rectification in the presence of $Mg^{2+}$ (*Figure 2A*). $Mg^{2+}$ thus appears to 'activate' G644P, by primarily increasing its conductance but not its open probability

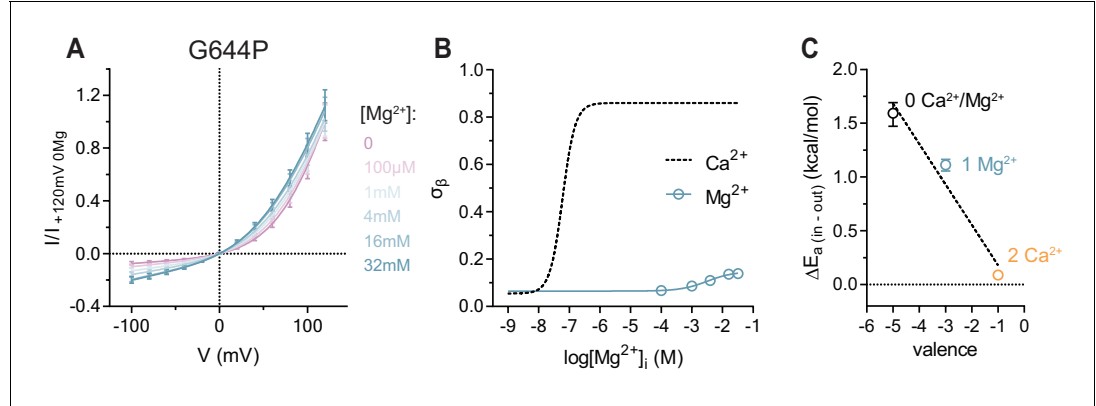

**Figure 2.** Conduction properties of the constitutively active mutant G644P in the presence of intracellular $Mg^{2+}$. (**A**) Instantaneous I-V relations of the constitutively active mutant G644P from pre-pulses at 80 mV in the presence of the indicated intracellular $Mg^{2+}$ concentrations. Solid lines are fits to *Equation 2*. Data were normalized to the fitted amplitude factor (*A* in *Equation 2*) and were subsequently normalized to the current amplitude at 120 mV at zero $Mg^{2+}$ (I/I $_{+120mV\ 0Mg}$). Data are mean values of normalized I-V plots from 9-11 individual patches, errors are s.e.m. (**B**) Relative rate of barrier crossing at the intracellular pore entrance ($\sigma_\beta$) as a function of intracellular $Mg^{2+}$ concentration. Solid line is a fit to the Hill equation for one binding site. The relation with $Ca^{2+}$ is shown as dashed line for comparison. Data are best-fit values and errors are 95% confidence intervals. (**C**) Experimental relationship between the assumed net charge of the $Ca^{2+}$ binding site (valence) and anion conduction energetics ($\Delta E_{a\ (in-out)}$). Data are transforms of $\sigma_\beta$, using *Equation 3*, at zero and saturating $Mg^{2+}$ and $Ca^{2+}$ concentrations. Dashed Line is a fit to *Equation 4*. The relative permittivity ($\varepsilon_r$) for occupancy by divalent cations was estimated to be 64.8 ± 35.1.

DOI: https://doi.org/10.7554/eLife.39122.006

The following figure supplements are available for figure 2:

**Figure supplement 1.** $Mg^{2+}$ concentration-response relations of WT and G644P.

DOI: https://doi.org/10.7554/eLife.39122.007

**Figure supplement 2.** Conduction properties of G644P in the presence of intracellular $Gd^{3+}$.

DOI: https://doi.org/10.7554/eLife.39122.008

(*Figure 2—figure supplement 1B and C*). Although we cannot exclude a non-specific effect of $Mg^{2+}$ due to the screening of surface charges, this is unlikely, as alanine mutations of positive charges in the wide vestibule have previously been shown to exert little effect on conduction (*Paulino et al., 2017b*). Interestingly, although $Mg^{2+}$ exerts a qualitatively similar effect on G644P as $Ca^{2+}$, the current remains rectifying even at saturating concentrations (*Figure 2B*). Moreover, in contrast to $Ca^{2+}$, the activation by $Mg^{2+}$ proceeds with a Hill coefficient close to unity and a maximum conduction rate at the intracellular entrance ($\sigma_\beta$) that saturates at an intermediate value, suggesting that the binding site is likely occupied by a single $Mg^{2+}$ ion (*Figure 2B and C*). Taken together, our data show three discrete levels of conductance that correspond to the possible occupancies of the binding site with zero, one and two divalent cations bound (*Figure 2C*), consistent with a scenario where sequential $Ca^{2+}$ binding lowers the energy barriers for anion conduction in a stepwise manner, as predicted by a simple Coulombic interaction. The model also describes the effect upon addition of the trivalent cation $Gd^{3+}$, which strongly increases the conductance at saturating concentrations even beyond the level induced by $Ca^{2+}$. This is likely a consequence of the increased positive charge density in the binding site and is manifested in the observed inward rectification of currents (*Figure 2—figure supplement 2*).

## $Ca^{2+}$ increases anion conductance by an electrostatic mechanism

Our results suggest that $Ca^{2+}$ and other di- and trivalent cations gate anion conduction in both constitutively active mutants by an electrostatic mechanism that involves neutralization of the negative charges in the vacant binding site. To gain further insight into this process, we investigated the effect of mutations of residues at the $Ca^{2+}$ binding site introduced in the G644P background and initially focused on E654Q located on α6. Whereas on a WT background, E654Q exhibits the most severe phenotype among binding site mutants and does not show any activity even at high $Ca^{2+}$ concentrations (*Brunner et al., 2014*; *Lim et al., 2016*; *Tien et al., 2014*), we observed basal and outwardly-rectifying currents in the mutant G644P/E654Q (PQ) whose steady-state current increases in response to $Ca^{2+}$ addition (*Figure 3—figure supplement 1A*). We examined the instantaneous I-V relations at increasing $Ca^{2+}$ concentrations and found a moderate concentration-dependent reduction of rectification that eventually saturates (*Figure 3A*). The much lower conduction rate of the PQ mutant compared to G644P at saturating $Ca^{2+}$ concentrations and a Hill coefficient close to one both suggest that in this mutant the binding of a single calcium ion to the transmembrane site promotes anion permeation, although less efficiently than the binding of two calcium ions in G644P, similar to the effect of $Mg^{2+}$ (*Figures 2* and *3A*).

We extended our investigation on Glu 654 and mutated this residue to an arginine (PR) to add a further positive charge to the binding site. Similar to PQ, this mutant likely binds only one calcium ion as the Hill coefficient is close to unity and the I-V relation retains its strong rectification even at saturating $Ca^{2+}$ concentrations (*Figure 3B*, *Figure 3—figure supplement 1B*). However, consistent with the hypothesis that a more positive potential at the binding site lowers the energy barrier for anion permeation, the rate of chloride conduction at the intracellular entrance at saturating $Ca^{2+}$ concentrations is significantly higher than that observed for PQ (*Figure 3B*). Next, we removed an additional negative charge in the binding site of the PR mutant by exchanging Glu 702 with Gln (PR2Q), which, similar to the equivalent mutation on the WT background (*Brunner et al., 2014*), reduces the potency of $Ca^{2+}$ (*Figure 3C*, *Figure 3—figure supplement 1C*). As expected, the rate of chloride conduction at the intracellular pore entrance in the triple mutant PR2Q is increased compared to the double mutant PR both in the absence of $Ca^{2+}$ and at saturating $Ca^{2+}$ concentrations (*Figure 3C*). Together these observations consolidate the notion of the dependence of anion conduction on the electrostatic potential around the binding site.

For electrostatic interactions, we expect the contribution of individual charges in the binding site to conduction to be additive. This hypothesis was tested by analyzing the effect of the removal of a negative charge in the binding site mutant E705Q in the PQ (PQ5Q) and PR (PR5Q) backgrounds. At maximum $Ca^{2+}$ occupancy, E705Q increases the rate of anion conduction to the same extent in both PQ5Q and PR5Q irrespectively of its background (*Figure 3D,E*), which emphasizes the orthogonal contributions of individual mutations (*Figure 3F*). The same mutations also exhibit a similar degree of additivity in the $Ca^{2+}$-free state (*Figure 3G*), further supporting an electrostatic mechanism for the functional interaction between the $Ca^{2+}$ binding site and the anion conduction path.

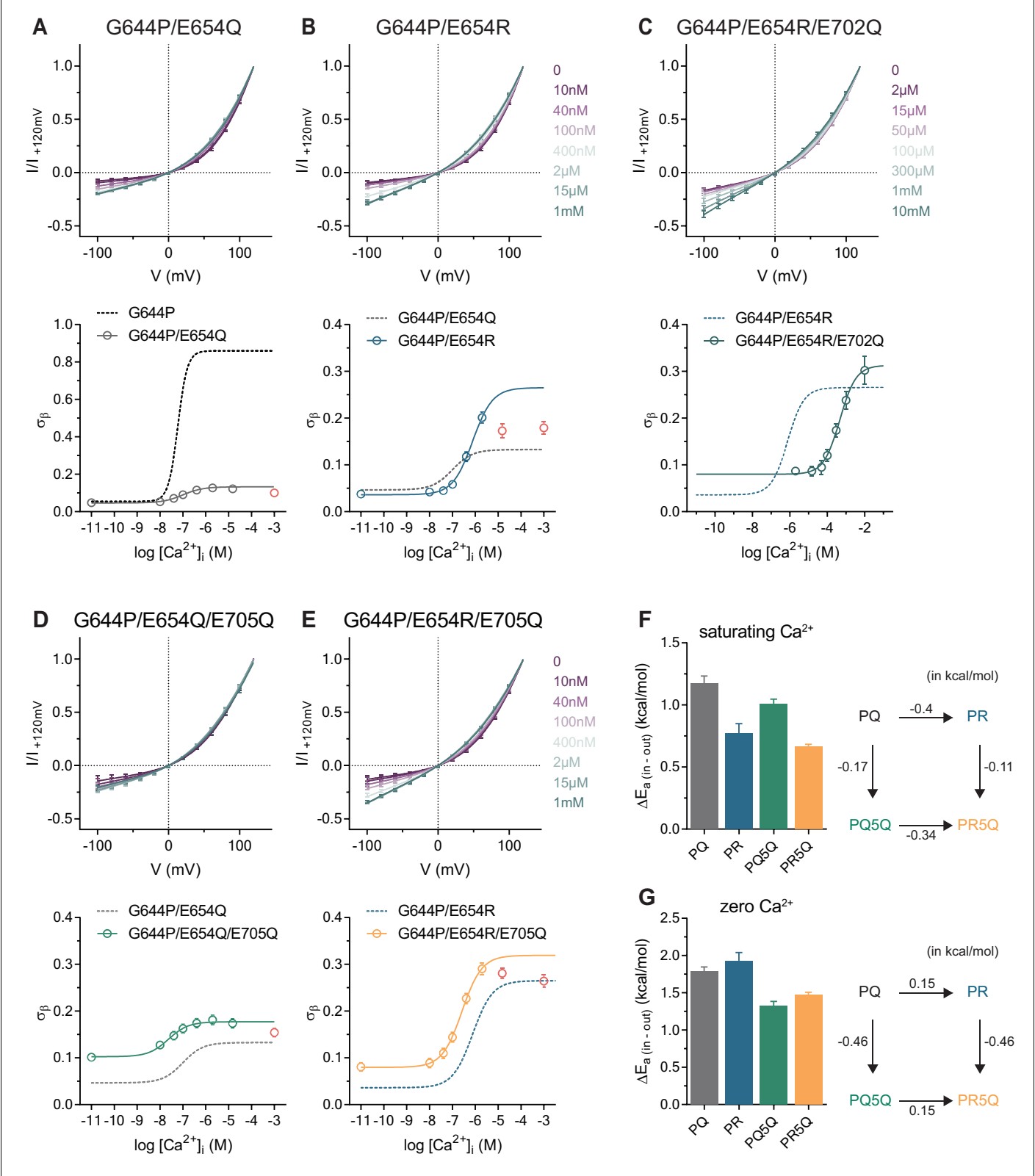

**Figure 3.** Conduction properties of the constitutively active mutant G644P with additional mutations at the $Ca^{2+}$ binding site. A-E, Top, instantaneous I-V relations from pre-pulses at 80 mV in the presence of the indicated intracellular $Ca^{2+}$ concentrations for mutants G644P/E654Q (A), G644P/E654R (B), G644P/E654R/E702Q (C), G644P/E654Q/E705Q (D) and G644P/E654R/E705Q (E). Solid lines are fits to *Equation 2*. Data were normalized to the current amplitude at +120 mV of each curve and are mean values of normalized I-V plots from 6, 7, 5–10, 5 and 7 individual patches respectively, errors
*Figure 3 continued on next page*

*Figure 3 continued*

are s.e.m. Bottom, relative rate of barrier crossing at the intracellular pore entrance ($\sigma_\beta$) as a function of intracellular $Ca^{2+}$ concentration for mutants G644P/E654Q (A), G644P/E654R (B) G644P/E654R/E702Q (C), G644P/E654Q/E705Q (D) and G644P/E654R/E705Q (E). Solid lines are fits to the Hill equation for one binding site. Data in red were omitted from the fit (see Materials and methods). Dashed lines are the relations of the indicated mutant shown for comparison. Data are best-fit values and errors are 95% confidence intervals. F-G, Left, relative activation energies at the intracellular pore entrance ($\Delta E_{a\ (in-out)}$) for the indicated mutants at saturating $Ca^{2+}$ concentrations (F) and zero $Ca^{2+}$ (G). Data are transforms of $\sigma_\beta$, using *Equation 3*, at saturating and zero $Ca^{2+}$ concentrations obtained from the fits shown in A-E. Right, energetic changes from the parent mutant G644P/E654Q (PQ) in a double-mutant cycle for the indicated mutants at saturating $Ca^{2+}$ concentrations (F) and zero $Ca^{2+}$ (G).
DOI: https://doi.org/10.7554/eLife.39122.009
The following figure supplement is available for figure 3:

**Figure supplement 1.** Concertation-response relations of mutants.
DOI: https://doi.org/10.7554/eLife.39122.010

## Electrostatic interactions are evident in calculations

To further strengthen our proposal of an electrostatic interaction between the bound $Ca^{2+}$ and permeating anions, we investigated the effect of the binding site mutations and the altered $Ca^{2+}$ occupancy on the electrostatic profile of the pore using Poisson-Boltzmann calculations on the $Ca^{2+}$-bound mouse TMEM16A structure (*Figure 4A*). Consistent with the quantal effect of $Ca^{2+}$ on anion conduction, removal of a single calcium ion from the binding site partially decreases the positive potential observed in the fully bound structure (*Figure 4B*). We also found that stepwise removal of charges in the binding site exerts progressively larger effects according to the number of charges neutralized (*Figure 4C*), as observed in our experiments. Although we cannot exclude the possibility that the detailed geometric arrangement of the $Ca^{2+}$ binding site and nearby helices might be influenced by the introduced mutations, local changes in backbone and sidechain conformations are unlikely to drastically affect the calculated electrostatic potential given the long-range nature of

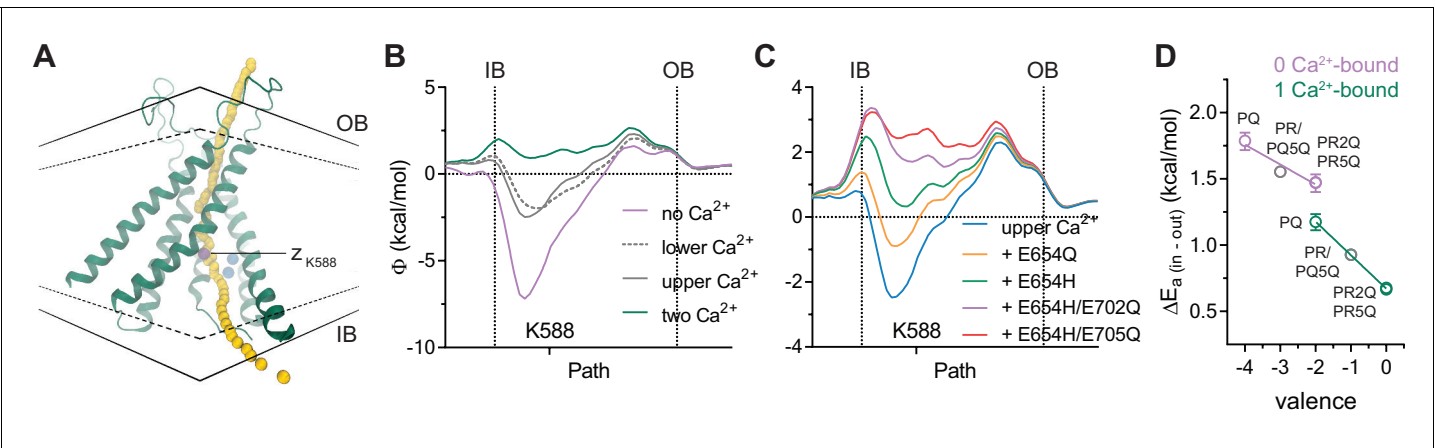

**Figure 4.** Electrostatic profiles. (A) System in which the electrostatic profile of the pore was calculated. Selected helices are shown and view is from within the membrane on one of the two pores in the dimeric protein. Black solid lines, outer- and innermost membrane boundaries (OB and IB, respectively); black dashed lines, boundaries of the hydrophobic core; blue spheres, $Ca^{2+}$ ions; yellow spheres, points at which the electrostatic potential ($\Phi$) was plotted. Magenta sphere corresponds to the z position of the nitrogen atom of Lys 588. B-C. Electrostatic potential along the pore in the $Ca^{2+}$-bound structure containing the indicated number of $Ca^{2+}$ ions (B) and carrying the indicated *in silico* mutations with only the upper $Ca^{2+}$ ion bound (C). Vertical dashed lines indicate the membrane boundaries. The position of Lys 588 at the intracellular pore entrance is indicated. (D) Experimental relationship between the assumed net charge of the $Ca^{2+}$ binding site (valence) and anion conduction energetics ($\Delta E_{a\ (in-out)}$). Data are transforms of $\sigma_\beta$, using *Equation 3*, at saturating and zero $Ca^{2+}$ concentrations for all the mutants on the G644P background (*Figure 2*). Lines are fits to *Equation 4*. The relative permittivity ($\varepsilon_r$) for the 0 $Ca^{2+}$-bound and 1 $Ca^{2+}$-bound states were estimated to be 162.7 ± 127.2 and 96.7 ± 6.3 respectively.
DOI: https://doi.org/10.7554/eLife.39122.011

Coulombic interactions. Because several mutants described above seem to bind only one calcium ion (*Figure 3* and *Figure 3—figure supplement 1*), it is possible to estimate from our experimental data the effective dielectric constant of the intracellular vestibule that propagates the electric field originating from the binding site to the pore. When the above mutants are analyzed collectively, we observed an inverse and linear relationship between the energy barrier for chloride conduction at the intracellular pore entrance and the valence of the binding site (*Figure 4D*). Assuming a Coulombic potential and using the distance measured from the $Ca^{2+}$-bound structure, we estimated an effective dielectric constant of around 160 in the vacant and 90 for the single $Ca^{2+}$-bound state. Although potentially inaccurate and physically unreasonable due to experimental limitations, these values indicate bulk solution-like aqueous properties that are consistent with the large water-filled intracellular vestibule observed in the $Ca^{2+}$-bound structure (*Figure 4D*). This also suggests that the vestibule is functionally shielded from the membrane.

## Bidirectional electrostatic interaction between the binding site and the intracellular pore entrance

In previous experiments, we have demonstrated that the bound $Ca^{2+}$ ions influence ion permeation by affecting the electrostatics at the narrow neck region via long-range Coulombic interactions.

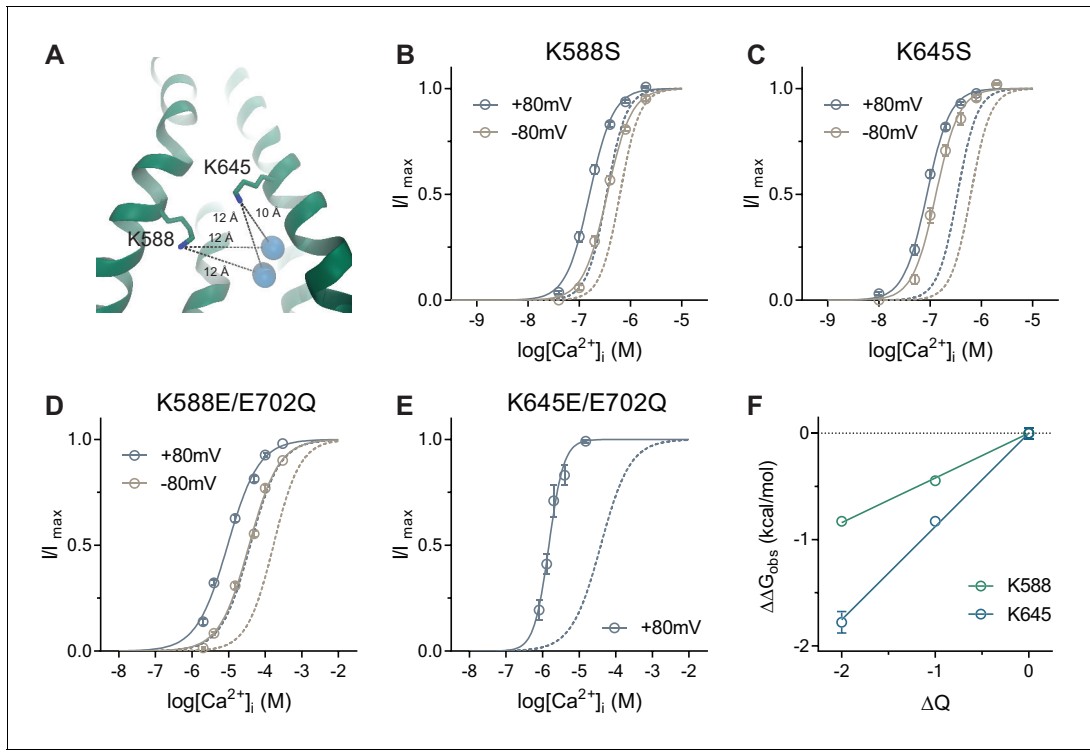

**Figure 5.** Activation properties of pore mutants. (**A**) Location of the pore residues Lys 588 and 645 relative to the $Ca^{2+}$ binding site (distances in Å). B-E. Concentration-response relations of K588S (**B**), K645S (**C**), K588E/E702Q (**D**) and K645E/E702Q (**E**) recorded at +/-80 mV using a rundown-correction protocol. Solid lines are fits to the Hill equation. Dashed lines are the relations of WT (**B–C**) and E702Q (**D–E**). Data are mean values of normalized concentration-response relations from 8-11, 7-8, 10 and 5-7 individual patches respectively, errors are s.e.m. (**F**) Relationship between the electrostatic potential of the intracellular pore entrance (ΔQ) and $Ca^{2+}$ binding energetics ($\Delta\Delta G_{obs}$) for the indicated residues. Data are transforms of $EC_{50\ mutant}/EC_{50\ background}$ at +80 mV using *Equation 6*. Solid lines are fits to *Equation 5*. The relative permittivity ($\varepsilon_r$) for Lys 588 and 645 were estimated to be 131.9 ± 12.3 and 71.2 ± 6.1 respectively.

DOI: https://doi.org/10.7554/eLife.39122.012

The following figure supplement is available for figure 5:

**Figure supplement 1.** Current traces of mutants at the neck region.
DOI: https://doi.org/10.7554/eLife.39122.013

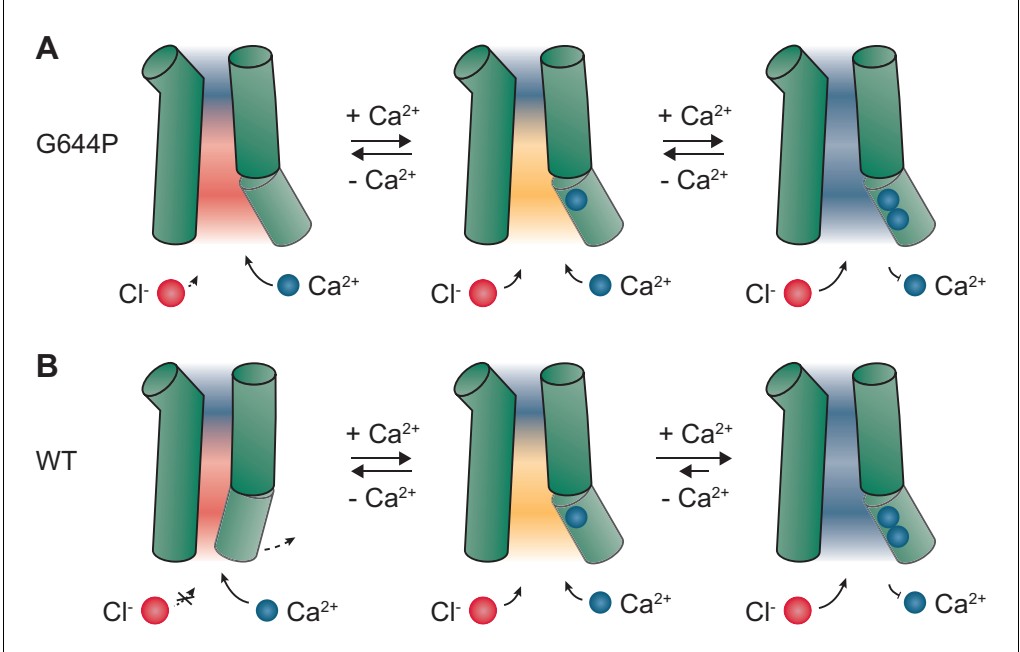

**Figure 6.** Mechanism. Schematic depiction of gating in G644P (**A**) and WT (**B**). In the apo state of the mutant G644P, the negative electrostatic potential strongly favors the access of $Ca^{2+}$ over $Cl^-$. In contrast, the positive electrostatic potential in the doubly occupied state strongly favors the access of $Cl^-$. A similar mechanism is expected to occur in WT, which requires $Ca^{2+}$ for the activation of α6 and the opening of a steric gate located above the intracellular vestibule. When activated, the major conducting state in WT is the open state with two $Ca^{2+}$ ions bound. Green cylinders, selected helices delimiting the pore; blue spheres, $Ca^{2+}$ ions; red spheres, $Cl^-$ ions. Red, orange and blue backgrounds depict negative, mildly negative and positive electrostatic potential in the pore.

DOI: https://doi.org/10.7554/eLife.39122.014

The following figure supplement is available for figure 6:

**Figure supplement 1.** Contribution of apo-, singly- and doubly occupied states in WT and G644P in a model of activation.

DOI: https://doi.org/10.7554/eLife.39122.015

Reciprocally, changes in the charge distribution at the neck should affect $Ca^{2+}$ binding. To test this hypothesis, we have investigated the effect of mutations of two residues, Lys 588 and Lys 645 (*Figure 5A*) located at the boundary between the intracellular vestibule and the neck, which lower the energy barrier for chloride conduction by contributing to the positive electrostatics of the pore (*Paulino et al., 2017b*). As expected, we found an increase in the potency of $Ca^{2+}$ in neutralizing mutations of the respective residues to serine (*Figure 5B,C*), which is further enhanced upon the reversal of the charge by mutations to glutamate (*Figure 5D,E*). Assuming that only the binding affinity of $Ca^{2+}$ is affected, we observed a linear relationship between the change in binding energy and the valence at the intracellular pore entrance (*Figure 5F*), with a stronger sensitivity for Lys 645, which is in closer proximity to the $Ca^{2+}$ binding site (*Figure 5A*) and thus might experience less solvent screening. In summary, the described effects further confirm the long-range interactions between the $Ca^{2+}$ binding site and the narrow neck region, which underlie the electrostatic control of ion permeation.

## Discussion

Our study demonstrates the strong electrostatic influence of bound $Ca^{2+}$ ions on anion conduction in the $Ca^{2+}$-activated $Cl^-$ channel TMEM16A. A direct effect of the ligand on the permeation properties of the open state is unique among ligand-gated ion channels and it is a consequence of the high positive charge density of $Ca^{2+}$ and the location of its binding site in the immediate vicinity of the

permeation path. The described effect is synergistic with a conformational change upon $Ca^{2+}$ binding that opens a steric gate in the narrow part of the pore (*Paulino et al., 2017a*). In the absence of $Ca^{2+}$, even in a sterically released conformation, which in this study was stabilized by mutations, an electrostatic gate imposed by the vacant binding site would impede the access of chloride ions to the intracellular vestibule (*Figure 6*). The binding of $Ca^{2+}$ releases this second gate by neutralizing the negative charges, enabling the open channel to conduct with full capacity (*Figure 6*). Our findings thus provide novel mechanistic insight into the regulation of anion permeation in $Ca^{2+}$-activated $Cl^-$ channels of the TMEM16 family and demonstrate that these channels are dually gated by both steric and electrostatic mechanisms. Moreover, they also provide a conceivable explanation for how the oppositely charged substrates, in this case permeating anions and $Ca^{2+}$ ions, access their respective site of action via the same intracellular vestibule (*Figure 6*). The observation that the conductance of the open channel can be electrostatically regulated by $Ca^{2+}$ in a stepwise manner also offers a unique opportunity to examine the relative contribution of open states with partial $Ca^{2+}$ occupancy during activation. The currents in such partially occupied states would be rectifying and thus be readily identifiable in the current-voltage relationships. In WT, the instantaneous I-V plots are all approximately linear and their shape is only mildly sensitive to $Ca^{2+}$ in the entire activation range (*Figure 1—figure supplement 2*). Since these plots reflect the weighted sum of the I-V relations of the open states with different $Ca^{2+}$ occupancy, our results indicate that, during activation, the major conducting state in WT is the open state with two $Ca^{2+}$ ions bound, whereas the contribution of open states with submaximal $Ca^{2+}$ occupancy is minimal (*Figure 1—figure supplement 2* and *Figure 6—figure supplement 1A,B*). A similar but less pronounced distribution of states is also evident in the constitutively active mutant G644P (*Figure 6—figure supplement 1C*), where the apo and the singly occupied channel make a larger contribution to conduction at low $Ca^{2+}$ concentrations. These observations thus provide direct evidence for the high cooperativity of TMEM16A activation.

Aside from their impact on membrane biophysics, our findings are particularly relevant for the development of allosteric activators of TMEM16A that may be beneficial for the treatment of obstructive airway diseases such as Cystic Fibrosis (*Li et al., 2017*; *Mall and Galietta, 2015*). Although we anticipate that small molecules have the potential to increase the low open probability of the channel at low $Ca^{2+}$ concentration, this might not suffice to promote robust $Cl^-$ efflux required to compensate for the loss of the chloride channel CFTR unless at the same time the electrostatic barrier at the intracellular pore entry is released, which would be the case for compounds with a positive net-charge or ones that at the same time increase the potency of $Ca^{2+}$.

## Materials and methods

### Key resources table

| Reagent type (species) or resource | Designation | Source or reference | Identifiers | Additional information |
|---|---|---|---|---|
| Gene (Mus musculus) | TMEM16A or Ano1 splice variant *ac* | DOI: 10.1038/nature13984 | UniProt identifier: Q8BHY3-1 | |
| Cell line (Homo sapiens) | HEK293T | ATCC | ATCC CRL-1573 | Obtained directly from ATCC; tested negative for myc oplasma contamination |
| Transfected construct (Mus musculus) | G644P | this paper | | generated using a modified QuikChange protocol as described in Methods |

*Continued on next page*

*Continued*

| Reagent type (species) or resource | Designation | Source or reference | Identifiers | Additional information |
|---|---|---|---|---|
| Transfected construct (Mus musculus) | Q649A | this paper | | as G644P |
| Transfected construct (Mus musculus) | G644P/E654Q | this paper | | as G644P |
| Transfected construct (Mus musculus) | G644P/E654R | this paper | | as G644P |
| Transfected construct (Mus musculus) | G644P/E654R/E702Q | this paper | | as G644P |
| Transfected construct (Mus musculus) | G644P/E654Q/E705Q | this paper | | as G644P |
| Transfected construct (Mus musculus) | G644P/E654R/E705Q | this paper | | as G644P |
| Transfected construct (Mus musculus) | K588S | this paper | | as G644P |
| Transfected construct (Mus musculus) | K645S | this paper | | as G644P |
| Transfected construct (Mus musculus) | K588E/E702Q | this paper | | as G644P |
| Transfected construct (Mus musculus) | K645E/E702Q | this paper | | as G644P |
| Recombinant DNA reagent | modified pcDNA3.1 vector | Invitrogen, DOI: 10.1085/jgp.201611650 | | bearing a 5' untranslated region (UTR) of hVEGF (from pcDNA4/HisMax; Invitrogen) |
| Software, algorithm | Clampex 10.6 | Molecular devices | | |
| Software, algorithm | Clampfit 10.6 | Molecular devices | | |
| Software, algorithm | Prism 5/6 | GraphPad | | |
| Software, algorithm | Excel | Microsoft | | |
| Software, algorithm | NumPy | http://www.numpy.org/ | | |
| Software, algorithm | CHARMM | https://www.charmm.org/ | | |
| Software, algorithm | VMD | https://www.ks.uiuc.edu/Research/vmd/ | | |

## Molecular biology and cell culture

HEK293T cells (ATCC CRL-1573) were maintained in Dulbecco's modified Eagle's medium (DMEM; Sigma-Aldrich) supplemented with 10 U/ml penicillin, 0.1 mg/ml streptomycin (Sigma-Aldrich), 2 mM L-glutamine (Sigma-Aldrich), and 10% FBS (Sigma-Aldrich) in a humidified atmosphere containing 5% $CO_2$ at 37 °C. HEK293T cells were transfected with 3 µg DNA per 6 cm Petri dish using the calcium phosphate co-precipitation method, and were used within 24–96 hr after transfection. Mutants

were generated with a modified QuikChange method (*Zheng et al., 2004*) using the wild-type mouse TMEM16A(*ac*) as the template. Double and triple mutants were generated sequentially. All constructs were verified by sequencing.

## Electrophysiology

Recordings were performed on inside-out patches excised from cells expressing the construct of interest. Recording pipettes were pulled from borosilicate glass capillaries (O.D. 1.5, I.D. 0.86, Sutter Instrument) and were fire-polished with a microforge (Narishige) before use. Pipette resistance was typically 3–8 MΩ when standard recording solutions were used. Seal resistance was typically 4 GΩ or higher. Voltage-clamp recordings were performed using Axopatch 200B and Digidata 1550 (Molecular devices). Analogue signals were filtered through the in-built 4-pole Bessel filter at 5 kHz and were digitized at 10–20 kHz. Data acquisition was performed using Clampex 10.6 (Molecular devices). Solution exchange was achieved using a double-barreled theta glass pipette mounted on an ultra-high speed piezo-driven stepper (Siskiyou). Liquid junction potential was found to be consistently negligible given the ionic composition of the solutions and was therefore not corrected. All experiments were performed at 20 °C.

Recordings were performed under symmetrical ionic conditions. Stock solution with $Ca^{2+}$-EGTA contained 150 mM NaCl, 5.99 mM $Ca(OH)_2$, 5 mM EGTA and 10 mM HEPES at pH 7.40. Stock solution with EGTA contained 150 mM NaCl, 5 mM EGTA and 10 mM HEPES at pH 7.40. Free $Ca^{2+}$ concentrations were adjusted by mixing the stock solutions at the required ratios, which were calculated using the WEBMAXC program (http://web.stanford.edu/~cpatton/webmaxcS.htm). Recording pipettes were filled with stock solution with $Ca^{2+}$-EGTA, which has a free $Ca^{2+}$ concentration of 1 mM.

For $Mg^{2+}$ experiments, stock solution with $Mg^{2+}$-EGTA contained 75 mM $MgCl_2$, 23 mM $(NMDG)_2SO_4$, 5 mM EGTA and 10 mM HEPES at pH 7.40. Free $Mg^{2+}$ concentrations were adjusted by mixing the stock solutions with $Mg^{2+}$-EGTA and EGTA at the required ratios. For $Gd^{3+}$ experiments, the final $Gd^{3+}$ concentrations were reconstituted from a 1 M $GdCl_3$ stock in a solution containing 150 mM NaCl, 10 mM HEPES at pH 7.40 with no added $Ca^{2+}$ and metal chelators. In all cases, solutions were prepared in ultrapure molecular biology grade water having a resistivity of 15–18 MΩ cm at 25 °C (ELGA or Millipore).

Concentration-response relations were constructed from steady-state current responses recorded using a rundown-correction protocol as described previously (*Lim et al., 2016*; *Paulino et al., 2017a*). For instantaneous I-V relations, current responses were measured at the time points (within milliseconds) where the capacitive current at 0 mV, the membrane potential at which the current reverses, has decayed. The sequence of the applied voltage steps was from −100 to +120 mV to minimize the effect of rundown on the smaller inward current (*Figure 1—figure supplements 1–3*, *Figure 2—figure supplement 1*). In a typical recording, the magnitude of the steady-state current at +80 mV would have decayed irreversibly by 10–15% by the time the test pulse at +120 mV is applied. To correct for this effect, the I-V plots were normalized to the current amplitude of the prepulses (*Paulino et al., 2017b*) before their normalization ($I/I_{+120}$). When used judiciously, this procedure allows one to recover the true I-V relation and to correct for current fluctuation. Before and after each recording at the $Ca^{2+}$ concentration of interest, we also recorded in zero $Ca^{2+}$ to ensure that the quality of the patch did not deteriorate over time. Background subtraction was performed for constructs that do not display visible basal activity but was not possible for the mutants that do. In all cases, leaky patches were discarded.

A similar procedure was used for experiments with $Mg^{2+}$ and $Gd^{3+}$. In contrast to the divalent cations $Ca^{2+}$ and $Mg^{2+}$, $Gd^{3+}$ binding appears to be biphasic consisting of a reversible phase and an irreversible component that both affect the I-V relation of the instantaneous current (Figure 2—figure supplement 2A). In order to titrate the second reversible site, we first incubated the patch with a submaximal concentration of $Gd^{3+}$ (0.4 mM) to saturate the irreversible binding site. We then recorded at the test concentration of $Gd^{3+}$ to obtain the corresponding instantaneous I-V relations (*Figure 2—figure supplement 2B,C*).

## Data analysis

Concentration-response data were fitted to the Hill equation

$$\frac{I}{I_{max}} = \frac{1}{1 + 10^{(logEC_{50} - log[Ca^{2+}])h}}$$

where $I/I_{max}$ is the normalized current, $EC_{50}$ is the concentration at which $I/I_{max}$ equals 0.5 and $h$ is the Hill coefficient. For the case of one binding site, h was set to equal 1. The voltage dependence of $EC_{50}$ was fitted to

$$logEC_{50} = logEC_{50(0)} - \frac{1}{2.303}\frac{z_{Ca}f_V VF}{RT} \qquad (1)$$

where $z_{Ca}$ is the valence of $Ca^{2+}$, $f_V$ is the electrical distance, $V$ is the membrane potential, $R$, $T$ and $F$ have their usual meanings, and $EC_{50(0)}$ is the value of $EC_{50}$ when $V = 0$.

I-V data were fitted to a minimal permeation model that accounts for the most fundamental biophysical behavior of mouse TMEM16A as described previously (*Läuger, 1973*; *Paulino et al., 2017b*),

$$I = zFAe^{\frac{zFV}{2nRT}}\frac{c_i - c_o e^{\frac{zFV}{RT}}}{e^{-zFV\frac{n-1}{nRT}} + \left(\frac{1}{\sigma_h}\right)\frac{1 - e^{-zFV\frac{n-2}{nRT}}}{\frac{zFV}{enRT} - 1} + \frac{1}{\sigma_\beta}} \qquad (2)$$

where $I$ is the current, $n$ is the number of barriers, $c_i$ and $c_o$ are the intracellular and extracellular concentrations of the charge carrier, $z$ is the valence of $Cl^-$ and $V$, $R$, $T$ and $F$ are defined as above. $A = \beta_0 v$ is a proportionality factor where $\beta_0$ is the value of $\beta$ when $V = 0$ and $v$ is a proportionality coefficient that has a dimension of volume. $\sigma_h$ and $\sigma_\beta$ are respectively the rate of barrier crossing at the middle and the innermost barriers relative to that at the outermost barrier ($\beta$),

$$\sigma_h = \frac{h_V}{\beta_V} = \frac{h_0 e^{\frac{zFV}{2nRT}}}{\beta_0 e^{\frac{zFV}{2nRT}}} = \frac{h_0}{\beta_0}$$

$$\sigma_\beta = \frac{\delta_V}{\beta_V} = \frac{\delta_0 e^{\frac{zFV}{2nRT}}}{\beta_0 e^{\frac{zFV}{2nRT}}} = \frac{\delta_0}{\beta_0}$$

where $h_0$ is the rate of barrier crossing at the middle barrier and $\delta_0$ at the innermost barrier in the absence of voltage. Thus, for a linear voltage drop as is the case in our model, $\sigma_h$ and $\sigma_\beta$ are intrinsically voltage-independent and represent the relative rates when $V = 0$. $\sigma_\beta$ appears to be voltage-dependent in *Figure 1G* because $\sigma_\beta$ is a function of $Ca^{2+}$ binding whose $EC_{50}$ exhibits voltage dependence.

In this model, current rectification is governed by the two parameters $\sigma_h$ and $\sigma_\beta$. Values of $\sigma_h$ and $\sigma_\beta$ that alter the symmetry of the energy profile (as exemplified in Figure 1—figure supplement 1A ) results in asymmetric I-V relations depending on the directionality of symmetry mismatch (*Paulino et al., 2017b*). We have previously shown that the model describes the behavior of WT best when the number of barriers $n$ equals 3 (*Paulino et al., 2017b*), which was used as a fixed parameter. This leaves $\sigma_h$, $\sigma_\beta$ and the amplitude factor $A$ as the only free parameters to be fitted. In this study, we focused on $\sigma_\beta$ because its best-fit value is generally better defined and it increases monotonically with the asymmetry of the I-V relation (rectification index (RI), $I_{-100/+120}$), which allows a straightforward comparison with the I-V data.

The $Ca^{2+}$ dependence of $\sigma_\beta$ and $\sigma_h$ was fitted to

$$\sigma_i = \sigma_{i(min)} + \frac{\sigma_{i(max)} - \sigma_{i(min)}}{1 + 10^{(logEC_{50} - log[Ca^{2+}])h}}$$

where the subscript $i$ indicates $\beta$ or $h$. We observed for mutants G644P/E654Q, G644P/E654R, G644P/E654Q/E705Q and G644P/E654R/E705Q a low affinity decay of $\sigma_\beta$ at high $Ca^{2+}$ concentrations (*Figure 3A, B, D, E*). Since this decay might be related to processes not intrinsic to the transmembrane binding site (*Lim et al., 2016*), only the high affinity phase was analyzed. The best-fit values of $\sigma_\beta$ at zero and saturating $Ca^{2+}$ concentrations were used to calculate $\Delta E_{a(in-out)}$, the difference between the activation energy at the innermost barrier relative to that of the outermost, using

$$\Delta E_{a\,(in-out)} = -RTln\sigma_\beta \tag{3}$$

The relationship between $\Delta E_{a\,(in-out)}$ and the valence of the binding site ($z_{bs}$) at different discrete $Ca^{2+}$ occupancies (0, 1 or 2) was fitted to the Coulombic potential

$$\Delta E_{a\,(in-out)} = \frac{-N_A z_{bs} q^2}{4\pi\varepsilon_0\varepsilon_r r} + \Delta E_{a\,(in-out)\,z_{bs}=0} \tag{4}$$

where $N_A$ is the Avogadro's number, $q$ is the electronic charge, $\varepsilon_0$ is the permittivity of vacuum, $\varepsilon_r$ is the relative permittivity of the medium, and $r$ is the distance between the binding site and the location of the innermost barrier. $z_{bs}$ was considered 0 when the five acidic residues (Glu 654, 702, 705, 734 and Asp 738) in the binding site are neutralized. $\varepsilon_r$ was estimated by setting $r$ to the measured distance between the center of the two calcium ions in the binding site and the sidechain oxygen atom of Ser 592 ($r$= 13.6 Å) in the structure of mouse TMEM16A in the $Ca^{2+}$-bound state.

An analogous expression was used to account for the changes in the energy of $Ca^{2+}$ binding as a function of valence of the intracellular pore entrance,

$$\Delta\Delta G_{obs} = \frac{N_A \Delta z_{pore} z_{Ca} q^2}{4\pi\varepsilon_0\varepsilon_r r} + \Delta\Delta G_{obs\,\Delta z_{pore}=0} \tag{5}$$

where $\Delta\Delta G_{obs}$ is the observed change in the free energy of $Ca^{2+}$ binding when only affinity is affected, $z_{pore}$ is the valence at the intracellular pore entrance and $z_{Ca}$ is the valence of $Ca^{2+}$. $z_{pore}$ was considered 0 for the WT protein. $\varepsilon_r$ was estimated by setting $r$ to the measured distance between the center of the two calcium ions in the binding site and the sidechain nitrogen atoms of Lys 588 or 645 ($r$= 11.9 Å or 10.6 Å) in the structure of mouse TMEM16A in the $Ca^{2+}$-bound state. $\Delta G_{obs}$ was calculated using

$$\Delta\Delta G_{obs} = RTln\frac{EC_{50}}{EC_{50(bg)}} \tag{6}$$

where $EC_{50(bg)}$ is the $EC_{50}$ of the background (WT or E702Q) on which the mutation of interest was constructed. We have constructed the K588E and K645E mutations on the E702Q background to improve the expression of the constructs. $\Delta G_{obs}$ is equivalent to $\Delta G_{binding}$ when the affinities of all binding reactions are affected equally without affecting conformational transitions, which is a general property of linked equilibria.

To estimate the contribution of open states with various $Ca^{2+}$ occupancy to the ensemble current, the I-V data for WT were fitted to a weighted sum of I-V curves corresponding to zero, one and two calcium ions bound at the binding site,

$$I_{total} = iI_{0Ca} + jI_{1Ca} + kI_{2Ca} \tag{7}$$

where $I_{0Ca}$, $I_{1Ca}$ and $I_{2Ca}$ are in the form of *Equation 2* with parameters from G644P at zero $Ca^{2+}$, G644P/E654Q at saturating $Ca^{2+}$ and G644P at saturating $Ca^{2+}$ respectively. Since WT does not display noticeable basal activity, the contribution of current corresponding to zero occupancy was considered negligible and the $iI_{0Ca}$ term was omitted.

Data analysis was performed using Clampfit 10.6 (Molecular devices), Excel (Microsoft), and Prism 5 and 6 (GraphPad). Numerical calculations were performed using the NumPy package (http://www.numpy.org/). Data are presented as mean ± s.e.m. Fitted parameters are plotted and are reported as best-fit value ± 95% confidence interval.

## Poisson-Boltzmann calculations

The electrostatic potential along the pore, identified using Hole (*Smart et al., 1996*), was calculated by solving the linearized Poisson-Boltzmann equation in CHARMM (*Im et al., 1998*; *Brooks et al., 1983*) on a 240 Å x 240 Å x 260 Å grid (1 Å grid spacing) followed by focusing on a 160 Å x 160 Å x 160 Å grid (0.5 Å grid spacing). Partial protein charges were derived from the CHARMM36 all-hydrogen atom force field (*MacKerell et al., 1998*). Hydrogen positions were generated in CHARMM. *In silico* mutagenesis was carried out in CHARMM. The protein was assigned a dielectric constant ($\varepsilon_r$) of 2. The membrane was represented as a 35-Å-thick slab ($\varepsilon_r$= 2). A 5-Å-thick slab was included on

each side of the membrane to account for the headgroup region ($\varepsilon_r$ = 30). The bulk solvent on either side of the membrane and the solvent-filled conduit were represented as an aqueous medium ($\varepsilon_r$ = 80) containing 150 mM mobile monovalent ions. Electrostatic calculations were carried out in the absence of an applied voltage. Although the structure of the protein used for calculations might not display a fully activated conformation, conformational changes are assumed to be small and would mainly affect the narrow neck region for which the dielectric constant is uncertain (*Paulino et al., 2017a*). The region most relevant for this study concerns the wide intracellular vestibule, which was assigned bulk-like dielectric properties. In calculations of mutants, positively charged residues were introduced as protonated histidines instead of arginines for modeling purposes.

## Acknowledgements

We thank all members of the Dutzler lab for help at various stages of the project.

## Additional information

### Funding

| Funder | Grant reference number | Author |
|---|---|---|
| H2020 European Research Council | ERC no 339116 AnoBest | Raimund Dutzler |

The funders had no role in study design, data collection and interpretation, or the decision to submit the work for publication.

### Author contributions

Andy KM Lam, Conceptualization, Data curation, Formal analysis, Validation, Investigation, Visualization, Methodology, Writing—original draft, Writing—review and editing; Raimund Dutzler, Conceptualization, Supervision, Funding acquisition, Validation, Project administration, Writing—review and editing

### Author ORCIDs

Andy KM Lam [ID] https://orcid.org/0000-0002-2983-3044
Raimund Dutzler [ID] http://orcid.org/0000-0002-2193-6129

### Decision letter and Author response

Decision letter https://doi.org/10.7554/eLife.39122.021
Author response https://doi.org/10.7554/eLife.39122.022

## Additional files

### Supplementary files

• Transparent reporting form
DOI: https://doi.org/10.7554/eLife.39122.016

### Data availability

All data generated or analysed during this study are included in the manuscript.

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
