## [Decision Letter]

Thank you for submitting your article "Ligand-dependent electrostatic control of anion access to the pore of the calcium-activated chloride channel TMEM16A" for consideration by *eLife*. Your article has been reviewed by three peer reviewers, and the evaluation has been overseen by a Reviewing Editor and Richard Aldrich as the Senior Editor. The following individuals involved in review of your submission have agreed to reveal their identity: Criss Hartzell (Reviewer#1) and Alessio Accardi (Reviewer #2).

The reviewers have discussed the reviews with one another and the Reviewing Editor has drafted this decision to help you prepare a revised submission.

Summary:

Lam and Dutzler discover that the negatively charged residues that form the Ca^2+^-binding sites act as an electrostatic barrier for the flow of anions once the pore has adopted the open conformation, thus acting as an additional activation gate in the absence of Ca^2+^. By introducing mutations in the Ca^2+^-sites in constitutively active mutants of TMEM16A and measuring their I-V relations at various Ca^2+^concentrations, the authors show that the binding of calcium reduces the two innermost energy barriers of an open channel through an electrostatic mechanism, allowing ions to flow. In the absence of Ca^2+^, the constitutively active mutants display pronounced outward rectification due to this electrostatic effect. Overall, these studies highlight a novel mechanism of regulation of ion permeation by a ligand. Nevertheless, the reviewers raise several concerns that should be addressed in the revised version.

Essential revisions:

1) I am not sure that I understand the permeation model. Why do the electrostatics of Ca^2+^ affect only inward and not outward current? The energy profile in Figure 1D could not lead to rectification, because the height of each energy barrier is the same whether the ion is coming from the inside or the outside. Outward rectification requires an energy well between the barriers *h* and *β* so that the height of *β* is greater from the out>in than the in>out direction. It would seem that the depth of this well would be significant considering the degree of rectification observed.

2) Furthermore, the second paragraph of the subsection “Data analysis” states that *σ_h_* and *σ_β_* are voltage-independent, but Figure 1G seems to show that *σ_β_* is voltage-dependent. The statement in the next paragraph is rather inscrutable, as is the subsequent explanation about why the "decay" was omitted from the fits. What does "inaccurate estimation of *σ_β_*" mean? It would be useful to know how the conclusions are affected by changing *n* in Equation 2. And why is *σ_h_* not shown beyond Figure 1?

3) Even though the G644P and Q649A mutations increase the baseline open probability of the channel, it is unclear whether the protein with any of these mutations still undergoes some gating-associated conformational changes. If this were the case, some of the effects of Ca^2+^-binding on the rectification of the mutant channels could arise from these conformational changes rather than the simple electrostatic influence of the site and the bound Ca^2+^ ions. To test for this possibility, the authors could use Mg^2+^ instead of Ca^2+^, as it is reported to compete with Ca^2+^ without activating the channel. The authors should test whether the presence of internal magnesium in the context of either of the constitutively active mutants has the same effect as Ca^2+^, as expected from their proposed mechanism. Although it is not necessary, I think it would be also interesting to test whether a trivalent cation could also bind and have a larger effect than Ca^2+^.

4) Large portions of the data shown depend on the use of a permeation model for analysis, together with the use of several equations for permeation and electrostatics. I think these play such a central role in the manuscript that it would make the reading easier if some of these models and equations were described with more detail in the main text, rather than the Materials and methods section.

---

## [Author Response]

Essential revisions:

1) I am not sure that I understand the permeation model. Why do the electrostatics of Ca^2+^ affect only inward and not outward current? The energy profile in Figure 1D could not lead to rectification, because the height of each energy barrier is the same whether the ion is coming from the inside or the outside. Outward rectification requires an energy well between the barriers h and β so that the height of β is greater from the out>in than the in>out direction. It would seem that the depth of this well would be significant considering the degree of rectification observed.

In our analysis, we rely on a simple rate theory model proposed by Peter Läuger (Lauger, 1973). We have explained this model in detail in our previous manuscript published in *eLife*, to which this work refers to (Paulino et al., 2017). In this model, ions surmount energy barriers in a pore that does not contain deep energy wells and that cannot be saturated. We think that such model is appropriate for our analysis since we have not found evidence for tight ion binding sites in our structures and we have measured a K_M_ of chloride conduction between 300-400 mM, thus underlining that there is no saturation in the applied conditions (Paulino et al., 2017). The energy profile displayed in Figure 1D or Figure 1—figure supplement 1 represents the potential of mean force for permeation in the absence of an applied electric field, where, in symmetric ion concentrations, the system is at equilibrium and there is no net flux in either direction. Upon application of a potential difference, the model predicts ionic currents. In such case, rectification is a consequence of the asymmetric free energy profile. The location of the barrier maximum governs the shape of the I-V relation, which becomes increasingly outwardly rectifying when the barrier maximum moves closer to the inner end. We note that the I-V relation is a function of both occupancy and rate of barrier crossing. Unlike the pseudo-ohmic cases, in case of low symmetric barriers, where the total occupancy remains approximately constant, ion depletion occurs in cases of an increased intracellular barrier at negative voltages where the channel is less conductive while ion accumulation occurs at positive voltages (Author response image 1). Both behaviors likely account for the resulting outward rectification and the general reduction in ion conductance. The latter corresponds to an effect that also affects the outward current (as indicated by the reduction of current at positive voltages in the I-V profiles in Author response image 1).

**Author response image 1. respfig1:** Model calculations based on the energy profile of G644P at zero Ca^2+^ (top) and a WT-like channel with a totally symmetric energy profile (bottom). Left, Ion occupancy in the model channel relative to that at the outermost wells (pn or p0) as a function of voltage at the indicated locations. Right, Shape of the I-V relation at steady state. At zero voltage, the relative occupancy is 1 for all cases because our model assumes no binding affinity when no voltage gradient is present. The curves on the left panels were calculated usingp1vc=ϕ+JvcβVpn−1vc=ϕn−1+JvcβV(ϕn−2+1σh(1−ϕn−21−ϕ))and those on the right panels usingI=zFJwhereJ=βVvc(1−ϕn)ϕn−1+1σhϕ(1−ϕn−21−ϕ)+1σβϕ=e−zFVnRTβV=β0ezFV2nRTunder symmetrical ionic conditions. The number of barriers n is 3 in this model. This has been determined to describe the conduction properties of mutants in our previous study (Paulino et al., 2017). βV is the voltage-dependent rate constant of the outermost barrier. The rate constant of the innermost barrier is defined as σββV and that of the middle barrier as σhβV where both fitted parameters σβ and σh are independent of voltage. pi’s are the probabilities of occupying the i^th^ wells (from p0 to pn). v is a proportionality factor that has a dimension of volume and may be interpreted as the hypothetical volume for outermost well at the channel entrance. V is the membrane potential, z is the valence of the ion, and R, T and F have their usual meanings.

2) Furthermore, the second paragraph of the subsection “Data analysis” states that σ_h_ and σ_β_ are voltage-independent, but Figure 1G seems to show that σ_β_ is voltage-dependent.

*σh* and *σβ* are both not intrinsically voltage-dependent as they denote the ratio of the rate constants in the case of a linear voltage drop (also see equation above). In Figure 1G they are only voltage-dependent because they change as a function of calcium binding and calcium binding itself is voltage-dependent. We have added the following sentences to our manuscript:

“σh and σβ are respectively the rate of barrier crossing at the middle and the innermost barriers relative to that at the outermost barrier (β),σβ=δVβV=δ0ezFV2nRTβ0ezFV2nRT=δ0β0where h0 is the rate of barrier crossing at the middle barrier and δ0 at the innermost barrier in the absence of voltage. Thus, for a linear voltage drop as is the case in our model, σh and σβ are intrinsically voltage-independent and represent the relative rates when V=0. σβ appears to be voltage-dependent in Figure 1G because σβ is a function of Ca^2+^ binding whose EC_50_ exhibits voltage dependence.”

The statement in the next paragraph is rather inscrutable, as is the subsequent explanation about why the "decay" was omitted from the fits. What does "inaccurate estimation of σ_β_" mean? It would be useful to know how the conclusions are affected by changing n in Equation 2.

We note that the decay phase in general does not follow the same trend as the high affinity phase and occurs an order of magnitude above the concentration range where activation takes place (Author response image 2). The decay might be related to additional processes conferred by Ca^2+^ not intrinsic to its high affinity binding, as we have observed in a previous study (Lim et al., 2016). As we are interested in the effect of calcium binding to the high affinity transmembrane site, the low affinity phase becomes irrelevant and was therefore omitted from analysis. Since the same treatment was applied consistently, this does not affect the outcome of the analysis. The number of barrier n is kept equal to that determined for the WT channel in our previous manuscript. This energy profile presents a minimal model required for a phenomenological description of ion permeation across a narrow neck region with two flanking barriers that account for the entry of ions into the neck whose heights are modulated by long-range electrostatics. It is reasonable to assume that the barrier heights are affected by the introduction of mutations, whereas the overall architecture of the energy profile remains similar.

We have added the following sentences:

“We have previously shown that the model describes the behavior of WT best when the number of barriers n equals 3 (Paulino et al., 2017b), which was used as a fixed parameter. This leaves σh, σβ and the amplitude factor Aas the only free parameters to be fitted.”

“We observed for mutants G644P/E654Q, G644P/E654R, G644P/E654Q/E705Q and G644P/E654R/E705Q a low affinity decay of σβ at high Ca^2+^ concentrations (Figure 3A, B, D, E). Since this decay might be related to processes not intrinsic to the transmembrane binding site (Lim et al., 2016), only the high affinity phase was analyzed.”

**Author response image 2. respfig2:** Relationship between experimental σβ and RI (I_-100/+120_) for the G644P family of constructs. Horizontal error bars indicate S.E.M and vertical error bars indicate the 95% confidence interval.

And why is σ_h_ not shown beyond Figure 1?

We have selected *σβ* for most of our graphs since it was determined with higher precision than *σh*. As shown in Figure 1E, the 95% confidence interval of *σh* is much wider than that of *σβ*, indicating that the best-fit value of *σh* is less well defined. In addition, the value of *σβ* increases monotonically (and almost linearly for the experimental range) with the rectification index (I_-100/+120_) (Author response image 2), which allows a straightforward comparison with the I-V data.

3) Even though the G644P and Q649A mutations increase the baseline open probability of the channel, it is unclear whether the protein with any of these mutations still undergoes some gating-associated conformational changes. If this were the case, some of the effects of Ca^2+^-binding on the rectification of the mutant channels could arise from these conformational changes rather than the simple electrostatic influence of the site and the bound Ca^2+^ ions. To test for this possibility, the authors could use Mg^2+^ instead of Ca^2+^, as it is reported to compete with Ca^2+^ without activating the channel. The authors should test whether the presence of internal magnesium in the context of either of the constitutively active mutants has the same effect as Ca^2+^, as expected from their proposed mechanism. Although it is not necessary, I think it would be also interesting to test whether a trivalent cation could also bind and have a larger effect than Ca^2+^.

We agree that gating-associated conformational changes may still exist. However, since for our analysis we characterize instantaneous currents following a voltage jump prior to any conformational changes, the I-V relations correspond to the open populations of the channel.

In our revised version, we have now investigated the effect of Mg^2+^ binding on the conductance. For that purpose, we have recorded data at varying Mg^2+^ concentration in the absence of Ca^2+^. Consistent with previous observations by Chen and colleagues (Ni et al., 2014), we found that Mg^2+^ binds with an apparent affinity of around 1-2 mM. Although Mg^2+^ addition does not lead to activation of WT, its binding to the mutant G644P lowers the energy barrier for anion conduction at the intracellular part of the pore, consistent with the presence of a single Mg^2+^ ion in the site (as judged by the remaining rectification at high Mg^2+^ concentration). This result thus further confirms our proposed mechanism on the influence of divalent cations in the transmembrane binding site on the energetics of anion conduction in the open channel. In another set of experiments, we have also investigated the effect of the trivalent metal ion Gd^3+^ and found that the application of this ion at high µM concentrations exerts an even stronger effect than Ca^2+^, leading to slight inward rectification of currents. This behavior is consistent with a larger increase in the positive charge density after the binding of two Gd^3+^ ions. However, in this case the effect was not completely reversible upon washout indicating an unmeasurably slow off-rate for one of the bound ions. It should also be noted that in the case of Gd^3+^, we could not buffer residual Ca^2+^ in the solutions by addition of EGTA and we can thus not exclude the presence of traces of Ca^2+^ in the solutions. In light of our new data, we have added another subsection to our manuscript “The effect of the competitive antagonist Mg^2+^, and the trivalent cation Gd^3+^ on conduction”) and an additional accompanying figure (see the new Figure 2 and Figure 2—figure supplements 1 and 2).

4) Large portions of the data shown depend on the use of a permeation model for analysis, together with the use of several equations for permeation and electrostatics. I think these play such a central role in the manuscript that it would make the reading easier if some of these models and equations were described with more detail in the main text, rather than the Materials and methods section.

We have added two sentences to the Results where we discuss general features of the model and extended its description in the methods. However, we have already explained the same model in great detail in our preceding study, which the current manuscript is referring to, and think that the link to this previous study would help interested readers to gain in-depth insight. We also think that a more extended description of the model in the main part of the manuscript might distract readers from the main findings of the work.

The following sentences were added to the Results section:

“The diffusion path does not contain deep energy wells and the model does not account for saturation of the pore, which is generally consistent with the high K_M_ for chloride conduction.”

“The rectification is a consequence of both pore occupancy and the rate of barrier crossing at the applied potential.”